# Effects of *Bradyrhizobium* Co-Inoculated with *Bacillus* and *Paenibacillus* on the Structure and Functional Genes of Soybean Rhizobacteria Community

**DOI:** 10.3390/genes13111922

**Published:** 2022-10-22

**Authors:** Pengfei Xing, Yubin Zhao, Dawei Guan, Li Li, Baisuo Zhao, Mingchao Ma, Xin Jiang, Changfu Tian, Fengming Cao, Jun Li

**Affiliations:** 1Institute of Agricultural Resources and Regional Planning, Chinese Academy of Agricultural Sciences, Beijing 100081, China; 2State Key Laboratory of Agrobiotechnology and Key Laboratory of Soil Microbiology, Ministry of Agriculture, College of Biological Sciences, China Agricultural University, Beijing 100094, China; 3Laboratory of Quality & Safety Risk Assessment for Microbial Products, Ministry of Agriculture, Beijing 100081, China

**Keywords:** rhizobium, bacillus, co-inoculation, rhizobacteria community, nitrogen cycle, phosphorus cycle, synergy function

## Abstract

Plant growth-promoting rhizobacteria (PGPR) are widely used to improve soil nutrients and promote plant growth and health. However, the growth-promoting effect of a single PGPR on plants is limited. Here, we evaluated the effect of applying rhizobium *Bradyrhizobium japonicum* 5038 (R5038) and two PGPR strains, *Bacillus aryabhattai* MB35-5 (BA) and *Paenibacillus mucilaginosus* 3016 (PM), alone or in different combinations on the soil properties and rhizosphere bacterial community composition of soybean (*Glycine max*). Additionally, metagenomic sequencing was performed to elucidate the profile of functional genes. Inoculation with compound microbial inoculant containing R5038 and BA (RB) significantly improved nodule nitrogenase activity and increased soil nitrogen content, and urease activity increased the abundance of the nitrogen cycle genes and *Betaproteobacteria* and *Chitinophagia* in the rhizosphere. In the treatment of inoculant-containing R5038 and PM (RP), significant changes were found for the abundance of *Deltaproteobacteria* and *Gemmatimonadetes* and the phosphorus cycle genes, and soil available phosphorus and phosphatase activity were increased. The RBP inoculants composed of three strains (R5038, BA and PM) significantly affected soybean biomass and the N and P contents of the rhizosphere. Compared with RB and RP, RBP consistently increased soybean nitrogen content, and dry weight. Overall, these results showed that several PGPR with different functions could be combined into composite bacterial inoculants, which coordinately modulate the rhizosphere microbial community structure and improve soybean growth.

## 1. Introduction

Soybean is a major source of high-quality proteins and edible oils for humans. The legume–rhizobia symbiotic nitrogen fixation has been of considerable significance for improving agricultural productivity. Plant growth-promoting rhizobacteria (PGPR) are particularly important in agricultural systems, and previous studies have reported that co-inoculation with PGPR and rhizobia could enhance rhizobia colonization of the rhizospheres of legumes, increase the number of nodules [1], improve the nitrogen fixation efficiency of symbionts [2,3] and promote plant nitrogen content [4]. Moreover, co-inoculation with PGPR and rhizobia can reduce the occurrence of soybean diseases, thus reducing pesticide use, improving yield, and protecting the environment [5,6,7]. Several studies have shown that the plant growth-promoting (PGP) effect of compound inoculants is better than that of a single inoculant [8,9,10]. However, the PGP mechanisms of compound inoculants have not been fully elucidated [11].

The rhizosphere contains a wide range of microorganisms and is an extremely complex environment. Application of PGPR strains to the rhizosphere may trigger interactions between the applied strains and other soil microorganisms, and these interactions play critical roles in regulating different chemical and biophysical processes in the rhizosphere, collectively promoting plant growth [12]. For example, some *Pseudomonas* and *Bacillus* strains can dissolve insoluble phosphate and potassium salts in soil [13,14,15], and some *Polymyxa* and *Rhizobium* strains can fix nitrogen from atmosphere [16], providing essential nutrients for crops. Additionally, PGPR can promote plant growth and enhance stress resistance by regulating plant hormone and enzyme activity [17,18]. Previous studies indicated that inoculation with PGPR capable of dissolving phosphorus, fixing nitrogen and producing indole acetic acid (IAA) improved the growth and performance of oats (*Avena sativa*), alfalfa (*Medicago sativa*) and cucumbers by increasing soil available nutrients [19]. Compared with single inoculation, co-inoculation with phosphorus-solubilizing bacteria (PSB) and nitrogen-fixing bacteria (NFB) exhibited significant PGP properties, including increasing soil nutrient contents and the accumulation of bioactive substances in leaves [20]. These results indicate that co-inoculation with different PGPR strains can increase soil available nutrients and promote root development and plant growth.

Similar to other crops, plant hormones produced by soybean rhizosphere PGPR are one of the main mechanisms affecting host plant development [21]. Puente et al. [22] reported that co-inoculation with *Bradyrhizobia* and *Azospirillum brasilense* improved soybean growth, which was attributed to IAA produced by PGPR. However, this finding cannot explain the synergistic effect of non-IAA-producing bacteria and rhizobia, and the interaction between soybean and PGPR may be affected by other factors, such as plant genotype, PGPR strain, inoculation dose, environmental conditions and soil microbial community structure. However, laboratory studies remain restricted to single or few species assemblies, with limited information on the interaction patterns and exogenous factors controlling the dynamics of natural microbial communities [23]. In the complex natural farmland soil ecosystems, the effects of PGPR inoculants may be unstable owing to the influence of multiple factors, such as climate, soil type and crops [24,25]. Moreover, studies on changes in soil microbial function after PGPR inoculation are limited. However, recent advances in metagenomics and high-throughput sequencing have promoted the understanding of changes in community structure caused by interactions between microorganisms [26,27]. Therefore, it is crucial to screen stable and efficient PGPR for specific environments and crop types through multiyear field experiments [28].

Particularly, it is important to screen PGPR strains with specific metabolisms and complementary functions when preparing a composite microbial inoculum. Inoculating the soybean rhizosphere with composite microbial inoculum is expected to improve interactions between microorganisms in the soil, with long-term and mutual benefits. In our previous work, one rhizobium, *B. japonicum* 5038 (R5038), and two PGPR, *B. aryabhattai* MB35-5 (BA) and *P. mucilaginosus* 3016 (PM), which were isolated from the rhizosphere of soybean, exhibited considerable PGP properties and improved nutrient use efficiency [29,30]. Here, three functional PGPR strains were applied sole or in different combinations to inoculate soybean (*Glycine max*) to further elucidate the interaction between PGPR and soybean. Additionally, the effect of the inoculants on the bacterial community composition and structure of the rhizosphere of soybean and soil properties was examined using metagenomic sequencing. Overall, we anticipate that the results of this study would help improve the understanding of the functions of compound microbial inoculants and rhizobium–soybean interaction.

## 2. Materials and Methods

### 2.1. Experiment Design

There were five treatments: (1) CK: noninoculated control; (2) R: inoculation with R5038 (*B. japonicum* 5038); (3) RB: inoculation with R5038 and BA (*B. aryabhattai* MB35-5); (4) RP: inoculation R5038 and PM (*P. mucilaginosus* 3016); and (5) RBP: inoculation with R5038, BA and PM. Each treatment had three replications. The characteristics of the three PGPR (including R5038) used in this study are summarized in Table 1. The pot experiment was conducted in a greenhouse in Beijing (39.96° N, 116.33° E), China. Fertilizer was not applied to any of the treatments. The soil type used in the experiment is classified as cinnamon, and its basic properties are as follows: organic matter, 19.7 g kg^−1^; total N, 0.97 g kg^−1^; available P, 14.8 mg kg^−1^; available K, 135.34 mg kg^−1^; and pH, 7.7. The soybean variety used is ‘Zhonghuang 39’, and three seeds were randomly sown per pot after watering. After germination, the plants were watered every 3 days. The greenhouse was maintained at 26 °C during the day and 22 °C at night, and stood about 14 h long in the sunshine.

### 2.2. PGPR Inoculant Preparation

The strain R5038 was cultured in YM liquid medium to the log phase (10^8–9^ CFU mL^−1^) at 28 °C for 4 days. BA was cultured in LB broth to the log phase (10^8–9^ CFU mL^−1^) at 30 °C for 12 h. PM was cultured in Silicate bacteria medium to the log phase (10^8–9^ CFU mL^−1^) at 30 °C for 3 days. For inoculation, three bacterial suspensions were mixed uniformly and then sprayed on the surface of the soybean seeds at the dose of about 2 mL·kg^−1^ (approximately 2.0 × 10^5^ cells per seed). Noninoculated control was similarly treated with sterile liquid medium. Thereafter, the seeds were air-dried in the shade before sowing.

### 2.3. Soil and Plant Samples Collection

Soil samples were collected from the rhizosphere of the soybeans at the flowering stage (60 days after germination), transferred to the lab in a cooling box (4 °C), and stored at −20 °C until DNA extraction. The other soil samples were air-dried in the shade and used to determine the soil physicochemical properties. Soil pH was determined using a glass combination electrode with soil:water of 1:1 [31]. The soil organic matter (SOM) and total nitrogen (TN) were determined according to Strickland and Sollins [32]. Soil KCl-extractable NO_3_^−^ and NH_4_^+^ were determined by extraction with 2 M KCl, steam distillation and titration [33]. Available P was analyzed by resin extraction following a protocol modified from Hedley and Stewart [34].

Soybean samples were collected to evaluate plant growth parameters, such as nodule number and shoot dry weight. Nitrogenase activity was determined immediately after nodule counting. Soybean shoot samples were dried in an oven at 70 °C until constant dry weight (DW) was obtained.

### 2.4. DNA Extraction and Metagenomics Sequencing

DNA was extracted from rhizosphere soil (0.25 g each) using PowerSoil DNA Isolation Kit (MoBio Laboratories, Inc., Carlsbad, CA, USA) according to the manufacturer’s instructions. The purity and quantity of the extracted DNA were assessed using 1% agarose gel electrophoresis and Qubit Flex Fluorometer (Thermo Fisher Scientific, Waltham, MA, USA), respectively. DNA extracts were stored at −20 °C. Library preparation and sequencing were conducted by Allwegene Tech Company (Beijing, China). Qualified DNA samples were randomly broken into small fragments of approximately 300 bp in length using Covaris S220 ultrasonic breaker, followed by pair-end repair, A-tail addition, sequencing adapter addition, purification and PCR amplification. Metagenomic libraries were prepared from 100 ng DNA per library using the NEBNext Ultra DNA Library Prep Kit (New England BioLabs, Ipswich, MA, USA), according to the manufacturer’s instructions. After the library was constructed, the library was diluted to 2 ng/µL, and the inserts of the libraries were checked using the Agilent 2100. The effective concentrations of the libraries (effective library concentration >3 nM), to ensure the quality of the libraries, was performed using the q-PCR method. High-quality libraries were pooled according to the requirements of effective concentration and target data volume, and sequencing was performed on the Illumina NovaSeq platform according to standard protocols. Data analysis was performed using open-source bioinformatic software and public databases.

### 2.5. Bioinformatics Analysis

Trimmomatic (v0.36) software was used for quality control of the raw data, including removing adapter sequences and low-quality reads. Thereafter, high-quality clean reads were aligned using the DIAMOND BLASTX algorithm, and species annotation was performed using NCBI nonredundant (nr) protein sequence and Kyoto Encyclopedia of Genes and Genomes (KEGG) databases. The sequencing data were assembled using MEGAHIT software (v1.0.6), and fragments less than 500 bp were filtered out. Gene prediction was performed using prodigal (v2.6.3) software and then screened for redundancy using CD-HIT (v4.8.1) software. The sequencing data were compared with the constructed nonredundant gene set using Bowtie (v1.1.2) software, and the relative abundance of single genes in different samples was calculated. Functional annotation of the nonredundant gene sets was performed using nr, Swiss-Prot, KEGG, COG/Kog, eggNOG, GO and Pfam databases.

### 2.6. Statistical Analysis

The experimental data for soil properties and soybean parameters were subjected to one-way ANOVA, followed by Duncan’s multiple range test (*p* < 0.05) for mean comparison, using SPSS statistics 19 (SPSS Inc., Chicago, IL, USA). Box-plots of soybean parameters were generated using GraphPad Prism 8. Heat maps of functional genes associated with N and P metabolism were performed on Tutools platform (http://www.cloudtutu.com, accessed on 6 September 2022), a free online data analysis website. Principal component analysis (PCA) and redundancy analysis (RDA) were performed for gene functional classification using Canoco 5.

## 3. Results

### 3.1. Parameters of Soil and Plant Growth

Pronounced differences in soil physicochemical properties were observed among the five treatments. Compared with CK, the soil physicochemical properties of inoculation treatments were kept in higher levels except for available potassium. The contents of total nitrogen, nitrate nitrogen and ammonium nitrogen in RB soil, 0.96 g kg^−1^, 14.08 mg kg^−1^ and 25.88 mg kg^−1^, respectively, are higher than those for other treatments (*p* < 0.05) (Table 2). Additionally, inoculation with RP and RBP significantly increased the available phosphorus content compared with the CK, with the RBP group having the highest value of 15.28 mg kg^−1^. Moreover, treatment with RBP significantly increased the soil organic matter content (18.20 g kg^−1^) compared with the other groups. Furthermore, soil urease activity was significantly higher in the RB and RBP treatments compared with the other three groups (*p* < 0.05). Similarly, inoculation with the PGPR significantly increased the phosphatase activity compared with the CK, with the RP group having the highest value. However, soil available potassium was not significantly affected by the treatments.

Additionally, soybean growth parameters were affected by the treatments (Figure 1). Inoculation with RBP slightly increased the number of nodules compared with the other treatments. Similarly, plants in the R, RB and RBP treatment groups had slightly higher nitrogenase activity than those in the CK and RP groups. The nitrogen content in the aboveground part of RBP soybean plants was the highest, followed by RB, R, RP and CK. Compared with CK, the soybean nitrogen content of RB and RP increased by 8.0% and 0.6%, respectively. Compared with RB and RP, the soybean nitrogen content of RBP increased by 4.1% and 11.6%, respectively. The shoot dry weight of RBP soybean plants was the highest, followed by RB, RP, R and CK. Compared with RB and RP, the shoot dry weight of RBP soybean plants increased by 7.8% and 7.1% (Figure 1a). In a word, RP could significantly increase the content of available P in rhizosphere soil; RB could significantly increase the content of soil N and had a significant effect on the accumulation of N in soybean plants; RBP could significantly increase the content of N and P and had the highest accumulation of N and biomass in soybean plants.

### 3.2. Rhizosphere Bacteria Abundance and Community Structure

More than 641 million clean reads were obtained from 15 samples, among which 47.15% were successfully annotated on the NCBI NR protein database. The relative abundances of the microbial genes were calculated, among which 98.49% were bacterial, 0.48% eukaryotic and 1.03% archaeal. At the class level, there were 12 groups with an average abundance >1% in all treatments. The most dominant classes include *Actinobacteria* (13%), *Alphaproteobacteria* (12%), *Betaproteobacteria* (7%), *Gammaproteobacteria* (5%), *Deltaproteobacteria* (5%), *Thermoleophilia* (2%), *Planctomycetia* (2%), *Gemmatimonadetes* (2%), *Bacilli* (1%), *Acidobacteriia* (1%), *Cytophagia* (1%) and *Chitinophagia* (1%) (Appendix A). The inoculation treatments had significant separation on the Simpson index with CK (*p* < 0.05), but there was no significant difference between treatments in terms of the Chao1, Shannon and ACE index (Appendix A). Compared with CK, inoculation with R5038 (R) had no significant change in soil bacterial community structure. The abundance of *Betaproteobacteria* and *Gemmatimonadetes* was significantly increased in inoculation treatments (RB, RP and RBP), while the abundance of *Thermoleophilia* decreased (Appendix A).

A Kruskal–Wallis test showed that there were significant differences in the bacterial abundance of 16 classes and 34 orders (Appendix A). RB inoculation had a higher effect on the rhizosphere bacterial community, significantly increasing the relative abundance of *Betaproteobacteria*, *Chitinophagia* and *Sphingobacteriia* but decreasing the abundance of *Actinobacteria* in class level (Appendix A). Moreover, Burkholderiales, Hyphomonadales, Bacteroidales, Chitinophagales and Sphingobacteriales were the most dominant bacterial orders in all treatments (Appendix A). Similarly, inoculation with RP increased the abundance of *Balneolia*, *Gemmatimonadetes* and *Saprospiria*; moreover, five orders, namely *Balneolales*, *Gemmatimonadales*, *Saprospirales*, *Pirellulales* and *Oligoflexales*, showed a highest abundance in RP. Furthermore, the rest orders with significant abundance differences in RBP had lower abundance than CK, RB and RP, except for *Iodidimonadales* (Appendix A). A heat map of the composition of bacterial communities showed that RB changed the bacterial community structure a lot, but treatment with RBP had a big similarity with RP and R in soil bacterial community structure (Figure 2).

### 3.3. Functional Diversity of Microbial Biomes Associated with the Rhizosphere of Soybean

KEGG pathway analysis of genes identified in soybean rhizosphere showed that more than half of the genes were enriched in the metabolic process. The significantly enriched pathways under cellular processes, environmental information processing, organismal systems and metabolism are shown in Appendix A. Principal component analysis (PCA) of the soil microbial metabolic function showed that the microbes formed distinct clusters according to the treatments, with PC1 and PC2 accounting for 15.83 and 12.19% of the total variation, respectively. The PCA models demonstrated the separation between RB, RP and CK, indicating that the functional genes were significantly influenced by different co-inoculations of PGPR (Figure 3a). Compared with control, the metabolic function of R had no significant differences; co-inoculation with two PGPR (RB and RP) resulted in significant changes in soybean rhizosphere microbial metabolic function; the PCA clustering of inoculation of RBP and RP were closer to that of CK and R treatments; and inoculation with RB showed separation of the rhizosphere microbial metabolic functions with CK and R. According to redundancy analysis (RDA), a total of 22.61% of the total variance was explained by the first two constrained axes of the RDA, the first axis explaining 12.54% and the second explaining 10.07% (*F* = 1.3, *p* = 0.012). All the environmental variables together explained 49.00% of the variation in microbial functions between samples. RDA analysis showed that RB rhizosphere microbial metabolism was highly correlated with soil N; RP rhizosphere microbial metabolism was highly correlated with soil P; and RBP rhizosphere microbial metabolism was correlated with soil N, P and organic matter (Figure 3b).

### 3.4. Genes Associated with Nitrogen Cycling and Phosphorus Cycling

The relative abundances of key genes associated with nitrogen cycling, such as nitrogen fixation (*nifKDH*), ammonia oxidation (*amoABC*, *hao*), nitrate reduction (*napAB*, *narGH*), nitrite reduction (*nirKS*), nitric oxide reduction (*norBC*), nitrous oxide reduction (*nosZ*) and nitrite ammonification (*nrfADH*, *nirBD*), were compared among the treatments. In the CK, relative abundance of genes related nitrogen metabolic were low, and only *nirBD*-related nitrite ammoniation and *nirS*-related nitrite reduction process were high. In the R rhizosphere soil, the ammonia oxidation process was the most active, the *amoAC* gene abundance was the highest and the nitrate reduction process dominated by *NarGH* and the main nitrogen fixation process of *nifHK* were also active. In the RP rhizosphere soil, nitrogen-related metabolic activities were weak, and only *nrfH*-related nitrite ammoniation and *nirK*-related nitrite reduction process were active. In RB rhizosphere soil, N-related metabolic activities were very active, and the abundance of most N-related genes was high; *norBC*-related nitric oxide reduction process and *napAB*-related nitrate reduction process were the most active; and *amoBC*-related ammonia oxidation process was also more active. In RBP rhizosphere soil, nitrogen-related metabolic activities were stronger than RP but weaker than RB. The abundance of *nrfD narGH*, *nirKS*, *norC* and *nifH* genes was also high (Figure 4a). Therefore, the nitrite ammonification, nitrate reduction, nitrite reduction, nitric oxide reduction and nitrogen fixation in RBP rhizosphere soil were also active. The results showed that, compared with other treatments, compound microbial inoculants RB could improve the genes related to N metabolism in soil, thus making the N metabolism process in rhizosphere soil more complex and diverse.

Furthermore, the expression levels of 22 genes associated with the phosphorus cycle were examined, including genes involved in mineralization (*aphA*, *phoADN*, *phn*, *ppx* and *ppa*), phosphorus transport and uptake (*ugp*, *pst*, *pit* and *pst*), regulatory systems (*phoBR*) and metabolism of phosphorous compounds (*ppk1*, *ppaC* and *ppgK*) (Figure 4b). In CK and R treatments, the abundance of genes related to P metabolism was generally low, and only the abundances of exopolyphosphatase gene *ppx* related to phosphorus dissolution and the nucleoside diphosphate kinase *ndk* related to polyphosphate degradation were higher. The abundances of alkaline phosphatase gene *phoADN*, phosphorus absorption regulating gene *phoBR* and polyphosphate synthesis-related gene *ppk1* were the highest in RB treatment. In the RP, the genes *acka*, *gltA*, *ldhA*, *aldb* and *ppc* related to organic acid synthesis; *gcd* and *phn* related to phosphorus dissolution; *ugp* and *pst* related to phosphate transport; and *ppaC* related to polyphosphate synthesis had the highest abundance. Among RBP, the abundances of *phn* related to phosphorus dissolution, *ppgK* related to polyphosphate degradation and phoBR related to phosphorus uptake regulation were the greatest. The results showed that the activity of RB phosphatase was higher, and RP was much stronger in dissolving phosphorus than others. In addition, although RB had more genes related to phosphorus metabolism, its functional redundancy was high, and it was not as comprehensive as RBP in phosphorus metabolism.

## 4. Discussion

### 4.1. Effects of PGPR Co-Inoculation on Microbial Community Structure in Soybean Rhizosphere Soils

Previous studies have shown that inoculating soybean with PGPR can significantly change the soil microbial community structure, regulate soil physical and chemical properties and promote soybean growth [35,36,37]. The effect of PGPR on soil microorganisms is influenced by the type of PGPR, inoculation method and inoculum size [2,38,39,40]. In this study, it was found that the soil microbial structure could be significantly altered by the inoculation of compound microbial inoculum in the form of seed coating under the condition of no fertilization.

Inoculation on soybean seed with R5038 and BA (RB) had a greater impact on the microbial community structure of rhizosphere soil. RB considerably affected the microbial community structure of the rhizosphere, as evidenced by a significant increase in the abundance of *Betaproteobacteria* (7.85%), *Chitinophages* (1.4%) and *Sphingolipids* (0.83%) and a decrease in *Actinomycetes* (11.63%) abundance compared with the other groups (Appendix A). *Betaproteobacteria* has been shown to improve ammonia oxidation and denitrification in the soil [41,42,43,44], as well as to promote the catabolism of complex organic matter [45], improve the utilization of soybean rhizosphere exudates by microorganisms [46] and enhance growth-promoting bacteria in soybean [47]. Similarly, R5038 and PM inoculum (RP) had a slight impact on the microbial diversity of rhizosphere soil, and only the abundance of *Deltaproteobacteria* (5.47%) and *Gemmatimonadetes* (1.72%) was increased (Appendix A). Among these *Deltaproteobacteria*, the sulfur-reducing bacteria *Desulfurellales* can utilize H_2_ and organic acids as electron donors and carbon sources for sulfate reduction metabolism [48,49,50]. Haeseleer et al. [51] found that *Gemmatimonadetes* has a series of cellulose hydrolase and laccase genes, which are involved in the lignin degradation process. Therefore, RB and RP can recruit microbial groups with different functions in the soybean rhizosphere, respectively, thus changing the rhizosphere microbial community structure.

Furthermore, RBP composed of R5038, BA and PM, had little change on rhizosphere soil microbial community structure, compared with RB and RP (Appendix A). The dominant bacterial groups recruited by RB and RP were not significant in RBP. Similar results were reported by Sarathambal et al. [52], who observed that the co-inoculation of *Rhizophagus* with *Bacillus megaterium* decreased the populations of *Conexibacter*, *Aneromyxobacter*, *Acidobacteria*, *Haliangium* and *Streptomyces*, which were abundant in single inoculation.

The differences of inoculations were more pronounced on bacterial communities recruited by RB and RP. This large shifts in these communities induced the increase in beneficial members and increased some essential metabolism pathways [53]. Moreover, the higher soil nitrogen and phosphorus and soybean plant dry weight exhibited by the RBP treatment also indicated that co-inoculation of multiple PGPR reduced the relative abundance of bacterial groups recruited by sole PGPR, possibly increased the metabolic efficiency of soil microbiota and promoted plant growth [54].

### 4.2. PGPR Co-Inoculation Significantly Shifted Soil Functional Profiles in Soybean Rhizosphere Soil

In the natural ecosystem, microorganisms live together in a complex network through reciprocity, competition and symbiotic interactions [55]. There are obvious differences in symbiotic patterns between bacterial communities. Functional complementary microorganisms have more connections and closer internal connections, then these connections playing a key role in the microecology [56]. In this process, the environment shapes the community composition through the interaction among microorganisms. Therefore, PGPR can affect other microorganisms and promote changes in the microbial community by changing the nutrients in the environments [57]. Our study showed that inoculating soybean seeds with compound microbial inoculants changed the microbial functional genes in the rhizosphere compared with CK. The compound microbial inoculants composed of BA or PM formed a significant functional cluster differentiation in soybean rhizosphere soil, indicating that BA and PM had a significant impact on the metabolic function of rhizosphere microorganisms.

The effect of the compound microbial inoculants RB, composed of R5038 and BA, on the metabolic function of N in soil is possibly achieved by BA-promoting nitrogen fixation in soybean root nodules. Liu et al. [58] found that both the formation of an infection thread and the development of nodules were related to the auxin signaling pathway. Sibponkrung et al. [6] also demonstrated that IAA affected soybean nodule development and nitrogen fixation efficiency through IAA-deficient mutants. BA, belonging to *B. aryabhattai*, promotes soybean growth and increases soil nitrogen through synthesizing IAA [59,60]. Then, in RB rhizosphere soil, the abundance of gene-related nitrification and denitrification (*norB*, *napAB*, *amoB*) were increased (Figure 4a), but the abundance of the gene-related nitrogen fixation (*nif*) was lower. The RB had the highest nitrate nitrogen and ammonium nitrogen content in the rhizosphere among all treatments. This result indicates that the nitrogen in RB rhizosphere mainly comes from nitrogen fixation by soybean nodules rather than nitrogen fixation from the air by soil microorganisms.

Furthermore, the compound microbial inoculants RP, composed of R and PM, also affected P metabolism in the rhizosphere. The compound microbial inoculants RP significantly increased the abundance of genes (*pflAD*, *ackA*, *ldhA*, *ppc*, *gltA*) related to organic acid synthesis in the rhizosphere of soybeans. These genes control the synthesis of organic acids by soil microorganisms, reduce soil pH, promote the mineralization of phosphorus compounds [61,62] and increase available phosphorus in the rhizosphere of RP. Thus, the abundance of genes (*phn*, *pho*, *pst*, *ugp*) related to P transformation in microorganisms was increased, and each P metabolism pathway was more active [63,64]. Therefore, compound microbial inoculants RP enhanced the dissolution of phosphorus in rhizosphere soil; increased the content of available P in soil, so as to make the metabolism of phosphorus in soil more active; and also provided P for soybeans.

The effect of the compound microbial inoculants RBP showed a different functional structure in rhizosphere soil. Inoculation with RBP increased the abundance of N reduction (*nirKS*, *nrfD*, *narGH*) and P metabolism-related genes (*phn*, *ppa*, *ppgK*, *pit*) and urease and phosphatase activities (Table 2). In addition, N metabolism and P metabolism are not two independent processes in this study. The nitrate reduction gene *narH* was positively correlated with the available P content in the soil (Figure 5a), while the phosphorase gene *phoADN*, pyruvate kinase gene *PK* and polyphosphate kinase gene *ppk1* were positively correlated with the N metabolism in the soil (Figure 5b). Biochemical processes of soil phosphorus are tightly coupled to the carbon and nitrogen cycle, which supports positive correlations among functional populations involved in carbon, nitrogen and phosphorus transformation [65]. This result indicated that nitrogen and phosphorus metabolisms in the soil are interrelated and promote each other, and the growth-promoting functions of the three different PGPR could be complementary or synergistic. Compared with RB and RP, RBP significantly increased the content of N and P in rhizosphere soil, promoted the nitrogen fixation capacity of root nodules, increased soybean biomass and had a better growth-promoting effect on soybean.

Therefore, inoculating PGPR into the soil will have a great impact on the species and abundance of indigenous microorganisms in the process of forming a new cooperative coexistence relationship with indigenous microorganisms. Compound microbial inoculants form a stable and persistent microbiome with versatile growth promoting functions in the rhizosphere through the complementarity of growth promoting functions.

## 5. Conclusions

In summary, inoculation with different compound PGPR enhanced the nitrogen fixation efficiency of soybean nodules and increased biomass yield; however, the treatments had different effects on the soil microbial community. Compound microbial inoculants composed of rhizobium, *B. japonicum* 5038 (R5038), and two PGPR, *B. aryabhattai* MB35-5 (BA) and *P. mucilaginosus* 3016 (PM), had altered the rhizosphere microbial community and metabolic function.

Inoculants RB and RP had stronger effects than RBP on the rhizosphere microbial community and functional diversity. RB improved the activity of nodule nitrogenase, increased the soil nitrogen content and regulated the nitrogen cycle. RP improved the soil phosphorus content and phosphatase activity and promoted the phosphorus cycle. Although RBP had a limited effect on microbial community structure and single N/P metabolism, it had the best effect on soil nitrogen, phosphorus contents and soybean biomass. Our results prove that a variety of PGPR with different functions can be combined into composite bacterial inoculants so as to have a better growth-promoting effect through the synergy of different growth-promoting functions.

## Figures and Tables

**Figure 1 genes-13-01922-f001:**
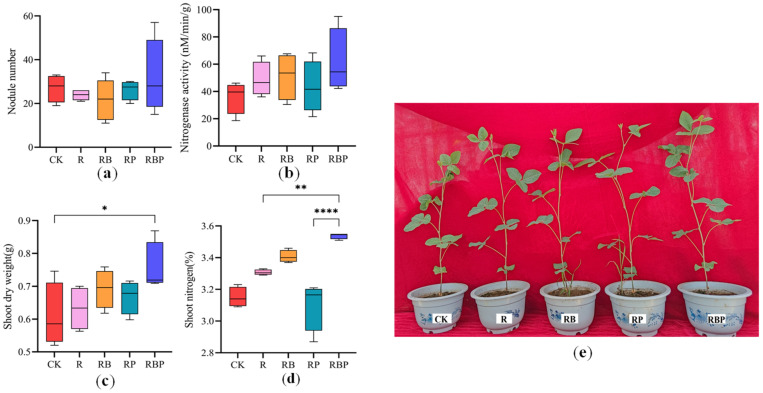
(**a**–**d**) Soybean growth parameters under different treatments (* 0.01 < *p* ≤ 0.05, ** 0.001 ≤ *p* ≤ 0.01, **** *p* ≤ 0.0001). (**e**) Soybean phenotypes under different treatments.

**Figure 2 genes-13-01922-f002:**
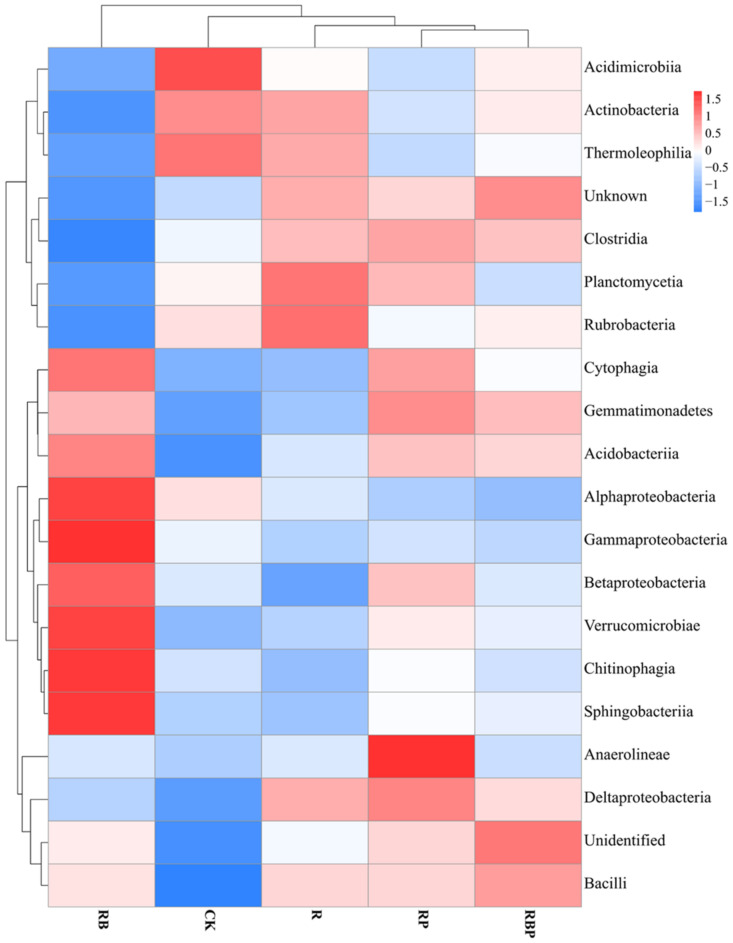
Cluster analysis of soil bacteria community (top 20 bacteria) at class level under different treatment.

**Figure 3 genes-13-01922-f003:**
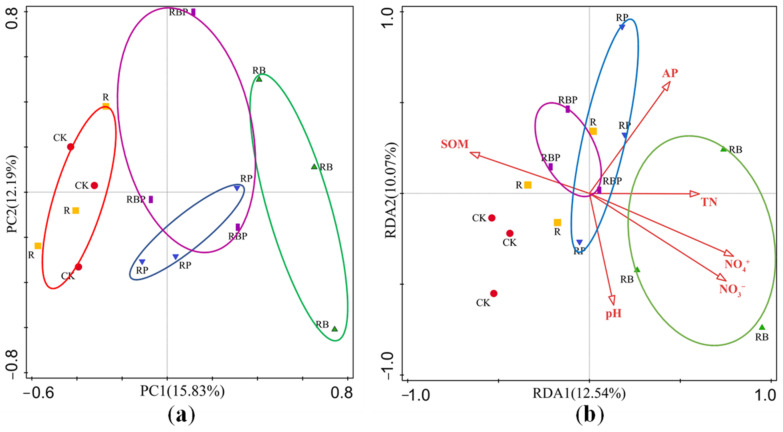
(**a**) Principal component analysis (PCA) between functional composition. (**b**) Redundancy analysis (RDA) between functional composition and soil properties, TN (total nitrogen), AP (available phosphorus) and SOM (soil organic matter).

**Figure 4 genes-13-01922-f004:**
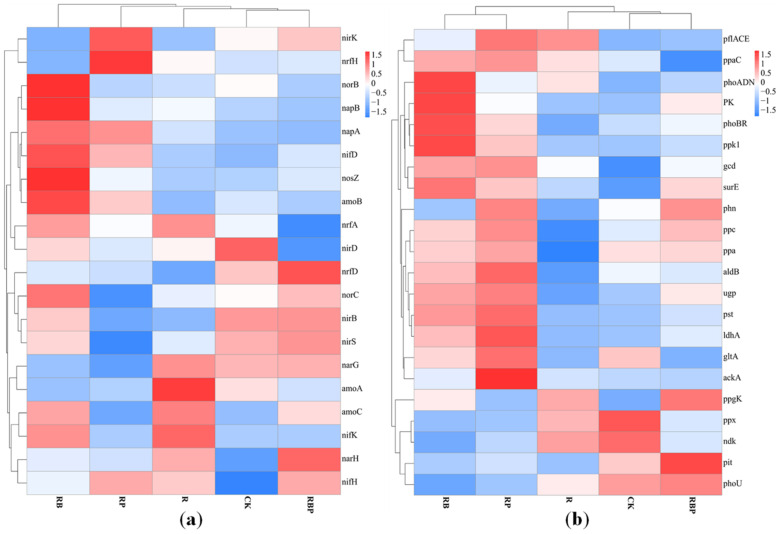
(**a**) Cluster analysis of genes related to N metabolism under different inoculation treatments. (**b**) Cluster analysis of genes related to P metabolism under different inoculation treatments.

**Figure 5 genes-13-01922-f005:**
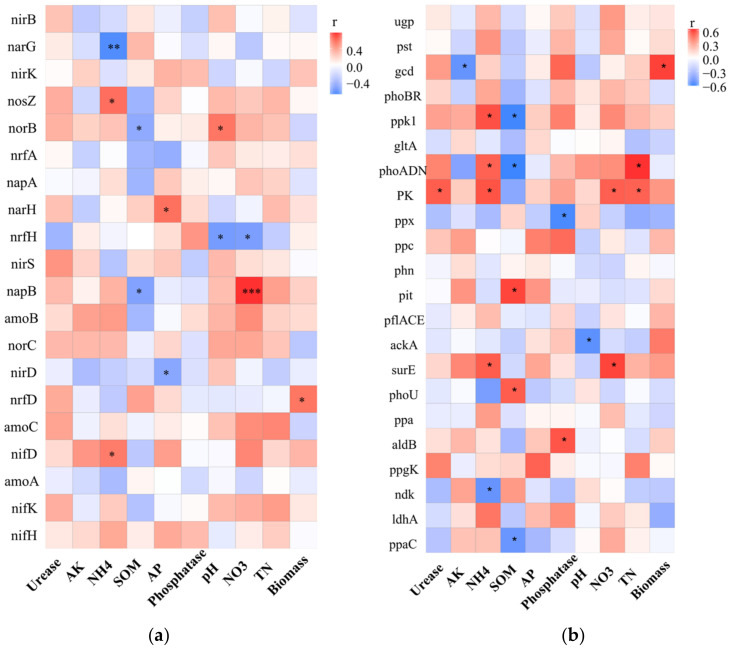
Spearman correlation coefficients between relative abundance of genes related to N metabolism (**a**) and P metabolism (**b**), * 0.01 < *p* ≤ 0.05, ** *p* ≤ 0.01, *** *p* ≤ 0.001.

**Table 1 genes-13-01922-t001:** Characteristics of the three strains.

Strains	Indolic Compounds (mg·L^−1^)	Inorganic Phosphate Dissolution	Potassium Dissolution	Nitrogen Fixation
R5038	−	−	−	+
BA	188.38 ± 6.25	−	−	−
PM	67.67 ± 2.62	+	+	−

**Table 2 genes-13-01922-t002:** Properties of rhizosphere soil.

T	Total Nitrogen (g/kg)	Nitrate Nitrogen (mg/kg)	Ammonium Nitrogen (mg/kg)	Available Phosphorus (mg/kg)	Available Potassium (mg/kg)	Soil Organic Matter (g/kg)	pH	Urease Activity (mg/kg)	Phosphatase Activity (IU)
CK	0.87 ± 0.016a	4.29 ± 0.53a	21.39 ± 0.95a	12.26 ± 0.84a	102.41 ± 1.09a	17.50 ± 0.16c	7.82 ± 0.042c	1094.27 ± 2.34ab	7.64 ± 0.12a
R	0.93 ± 0.005c	7.27 ± 2.28a	22.66 ± 0.57ab	13.51 ± 0.38ab	102.42 ± 1.30a	17.57 ± 0.19c	7.75 ± 0.025b	1105.15 ± 7.54b	8.58 ± 0.21b
RB	0.96 ± 0.005d	14.08 ± 3.30b	25.88 ± 1.24c	13.66 ± 1.48ab	102.35 ± 1.15a	16.50 ± 0.08a	7.85 ± 0.014c	1130.66 ± 3.64c	8.69 ± 0.32b
RP	0.90 ± 0.005b	5.18 ± 0.23a	24.00 ± 1.61bc	14.32 ± 1.30b	101.52 ± 1.85a	17.03 ± 0.19b	7.66 ± 0.026a	1091.01 ± 0.40a	9.63 ± 0.41c
RBP	0.94 ± 0.005c	7.38 ± 2.66a	22.86 ± 0.56ab	15.28 ± 0.22b	102.02 ± 0.38a	18.20 ± 0.16d	7.70 ± 0.025ab	1130.41 ± 0.75c	9.16 ± 0.18bc

Different letters in each column indicate significant difference among treatments (*p* < 0.05).

## Data Availability

The obtained sequences were submitted to the National Center for Biotechnology Information (NCBI) Sequence Read Archive (SRA) with BioProject number PRJNA892802. Other data that supports the findings of this study are available in the supplementary material of this article.

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
