# Peer review of "Effects of Bradyrhizobium Co-Inoculated with Bacillus and Paenibacillus on the Structure and Functional Genes of Soybean Rhizobacteria Community"

_genes, 2022, doi:10.3390/genes13111922_

Round 1
Reviewer 1 Report
This article represents a very well conducted study on the composition of the content of soil microbes. I consider that it is an original work that dimensions the knowledge of soil microbiology from its microbial composition, its gene expression alternatives, and the metabolic profile of its components.
It represents a study model that will allow, based on knowledge, the use of biological fertilization schemes that allow enriching agricultural soils and thereby achieving higher yields.
Author Response
Response to Reviewer 1 Comments
Dear reviewer:
Thanks very much for taking your time to review this manuscript. We really appreciate for your positive and constructive and Comments on our manuscript entitled “Effects of different Bacillus co-inoculated with Bradyrhizobium japonicum 5038 on the structure and functional diversity of soybean Rhizobacteria community” (genes-1952976)! We have tried our best to improve and made some changes in the manuscript in revisions mode.
Reviewer 2 Report
The manuscript entitled "Effects of different Bacillus co-inoculated with Bradyrhizobium japonicum 5038 on the structure and functional diversity of soybean Rhizobacteria community" is presenting a very interesting work concerning the synergic effects of Rhizobia and PGPR bacteria on soybean Rhizobia community.
Major points:
Study with five treatments on a non define soil with organica matter will lead as expected to non obvious analysis of result. What are the bradyrhizobia already in the soil in control experiment, and how competitive is B. japonicum 5038 in comparison with others strains. To my understanding, this is a huge problem for the complete study.
Then as expected in Figure 1, Control plants are fine and are developping roughly the same amount of nodules than inoculated plants with Bradyrhizobium (R). There is no effect on shoot nitrogen with or without inoculation. Then why to look then for synergic effect of PGPR if I don't even know the bacterial community in the soil. Are these two PGPR presents too, I have no clue about this question.
Figure 3: Repeat of three independent experiment are not consistent, then nothing can be obtain from this result. Please remove such data !
Figure 4: Should be present as supplementary data due to the lack of obvious interesting data.
Minors points:
Remove or change lines 3-94
Line 98: Why control is CK and not simply C
Lines 119-129: How many independant repeat ?
Line 142: Change for 2 ng/µl
Coherence between Title (line 204) and text lines 205-213 ???? What is exactly Rhizobacterial bacteria ? Must be defined !
The analysis of figure 2 is partial and do not help for understanding nearly insignificant results. Move Fig 2 in supplemental data.
Table S1 column H in chinese ????
Table S2 What are the difference between unindentified and unknow ?
Figure 7: Why this fig 7 appears here in discussion ?
Lines 409-415: What about the long terme effect of such bacterial treatment on long term soil ecological evolution after release from plants of B japonicum et PGPR.
Why to present in lines 419-422 the different strains ? Already done in lines 98-100.
